# Investigating the association between genetically proxied circulating levels of immune checkpoint proteins and cancer survival: protocol for a Mendelian randomisation analysis

Tessa Bate ![ORCID] ,[1,2] Richard M Martin,[1,2,3] James Yarmolinsky,[1,2,4] Philip C Haycock[1,2]

JY and PCH contributed equally.

For numbered affiliations see end of article.

**Correspondence to**
Dr Philip C Haycock;
philip.haycock@bristol.ac.uk

## ABSTRACT

**Introduction** Compared with the traditional drug development pathway, investigating alternative uses for existing drugs (ie, drug repurposing) requires substantially less time, cost and resources. Immune checkpoint inhibitors are licensed for the treatment of certain breast, colorectal, head and neck, lung and melanoma cancers. These drugs target immune checkpoint proteins to reduce the suppression of T cell activation by cancer cells. As T cell suppression is a hallmark of cancer common across anatomical sites, we hypothesise that immune checkpoint inhibitors could be repurposed for the treatment of additional cancers beyond the ones already indicated.

**Methods and analysis** We will use two-sample Mendelian randomisation to investigate the effect of genetically proxied levels of protein targets of two immune checkpoint inhibitors—programmed cell death protein 1 and programmed death ligand 1—on survival of seven cancer types (breast, colorectal, head and neck, lung, melanoma, ovarian and prostate). Summary genetic association data will be obtained from prior genome-wide association studies of circulating protein levels and cancer survival in populations of European ancestry. Various sensitivity analyses will be performed to examine the robustness of findings to potential violations of Mendelian randomisation assumptions, collider bias and the impact of alternative genetic instrument construction strategies. The impact of treatment history and tumour stage on the findings will also be investigated using summary-level and individual-level genetic data where available.

**Ethics and dissemination** No separate ethics approval will be required for these analyses as we will be using data from previously published genome-wide association studies which individually gained ethical approval and participant consent. Results from analyses will be submitted as an open-access peer-reviewed publication and statistical code will be made freely available on the completion of the analysis.

## INTRODUCTION

Drug repurposing is the use of approved drugs for another indication.[1 2] The traditional development and testing pathway of candidate drugs is expensive and time-consuming,

## STRENGTHS AND LIMITATIONS OF THIS STUDY

⇒ As germline genetic variants proxying circulating protein levels are randomly assorted at meiosis and fixed at conception, Mendelian randomisation analyses examining the effect of these proteins on cancer survival should be less prone to conventional issues of confounding and cannot be influenced by reverse causation bias.

⇒ The use of a two-sample Mendelian randomisation framework will permit us to leverage large-scale genetic association data from separate samples, thus enhancing statistical power and precision of estimates.

⇒ The generalisability of our findings to populations of non-European ancestry may be unclear.

⇒ Mendelian randomisation analysis can only evaluate the on-target effects of immune checkpoint inhibitors.

⇒ Sample sizes of genome-wide association studies of cancer survival are low in comparison to cancer risk, which will limit the statistical power of our analyses.

with an estimated cost of US$2–3 billion and 13 years of research on average required for a chemical compound to be approved for use in clinical practice.[3] In contrast, drugs that are tested for a repurposed use should already have demonstrated success in phase I trials for their original indication and thus their safety profiles for human use are known.[3–6] Consequently, clinical testing for a repurposed use of a drug can begin at phase II trials, reducing associated time and resource requirements.[3–6]

Despite advances in screening and treatment strategies, the number of people diagnosed with, and dying from cancer, continues to increase. Globally, there were estimated to be 19.3 million new cancer diagnoses and 10.0 million cancer deaths in 2020.[7] Seven cancer sites (breast, colorectal, head and

neck, lung, melanoma skin, ovarian and prostate cancer) were estimated to contribute to 48% of the incidence and 45% of mortality from all cancer sites globally in 2020.[7] In addition to the high burden of cancer, there are issues associated with currently available treatments such as development of resistance, severity of side effects and lack of efficacy in some individuals.[8] Identifying new strategies for the treatment of these high-burden cancers using drug repurposing could minimise the cost and patient involvement required for the assessment of their efficacy. Several drugs have been successfully repurposed for cancer treatment, including non-cancer drugs such as thalidomide which was originally developed to treat morning sickness in pregnancy but is now approved to treat multiple myeloma.[3 8 9]

Shared hallmarks of cancer common to different cancer sites represent an opportunity for drug repurposing using approved drugs which target these mechanisms across multiple sites.[3] One such hallmark is the avoidance of immune destruction, which can be suppressed using immune checkpoint inhibitors.[3 10 11] The first immune checkpoint inhibitor approved by the US Food and Drug Administration was the anti-cytotoxic T-lymphocyte associated protein 4 monoclonal antibody, ipilimumab, for the treatment of melanoma in 2011.[10] Following the approval of ipilimumab, several other immune checkpoint inhibitors have also been approved for a range of cancer indications.

Two examples of immune checkpoint proteins which have been successfully targeted in cancer treatment are programmed cell death protein 1 (PD-1) and programmed cell death ligand 1 (PD-L1).[10] The anti-PD-1 monoclonal antibodies include cemiplimab (Libtayo), dostarlimab (Jemperli), nivolumab (Opdivo) and pembrolizumab (Keytruda), and the anti-PD-L1 monoclonal antibodies include atezolizumab (Tecentriq), avelumab (Bavencio) and durvalumab (Imfinzi).[10 12] These seven immune checkpoint inhibitors have been approved by the Medicines and Healthcare products Regulatory Agency (MHRA) for specific cancer indications, including some

indications for the seven cancer types detailed previously (table 1, online supplemental table 1). Across anti-PD-1 immune checkpoint inhibitors, there are approved indications for the treatment of breast, colorectal, head and neck, and lung cancers and melanoma, while anti-PD-L1 immune checkpoint inhibitors have been approved for breast and lung cancer treatment[13–19] (table 1, online supplemental table 1).

The approvals for anti-PD-1 and anti-PD-L1 immune checkpoint inhibitors are not uniform across cancer sites, even for the drugs targeting the same immune checkpoint protein (table 1). However, this may be explained by an absence, rather than failure, of comparable clinical trials for certain drugs within these drug target categories. For example, pembrolizumab is the only anti-PD-1 immune checkpoint inhibitor approved for a breast cancer indication by the MHRA[16] (table 1), likely due to lack of complete late-stage clinical trials with large sample sizes investigating the efficacy of the other three anti-PD-1 immune checkpoint inhibitors.[20 21] Similarly, although nivolumab and pembrolizumab both have colorectal cancer indications, there are differences in the characteristics of the patient populations these drugs are approved to treat[15 16] (table 1, online supplemental table 1). Nivolumab is approved to treat patients with mismatch repair deficient (dMMR) or microsatellite instability high (MSI-H) colorectal cancer after chemotherapy as part of a combination therapy with ipilimumab[15] (online supplemental table 1). In contrast, pembrolizumab as monotherapy is approved to treat metastatic or unresectable dMMR/MSI-H colorectal cancer, the latter following previous treatment[16] (online supplemental table 1).

The approved indications of these immune checkpoint inhibitors are highly specific in many cases, particularly with respect to molecular tumour markers and treatment history. For example, durvalumab has been approved by the MHRA for the treatment of adult patients with locally advanced, unresectable non-small cell lung cancer if at least 1% of their tumour cells express PD-L1 and their disease did not advance after previous platinum-based

**Table 1** Medicines and Healthcare products Regulatory Agency (MHRA) indications of anti-programmed cell death protein 1 (PD-1) and anti-programmed cell death ligand 1 (PD-L1) monoclonal antibodies, obtained 25 March 2023

| Protein target | Immune checkpoint inhibitor | Cancer | | | | | | |
| | | Breast | Colorectal | Head and neck | Lung | Melanoma | Ovarian | Prostate |
|---|---|---|---|---|---|---|---|---|
| PD-1 | Cemiplimab[13] | – | – | – | A | – | – | – |
| | Dostarlimab[14] | – | – | – | – | – | – | – |
| | Nivolumab[15] | – | A | A | A | A | – | – |
| | Pembrolizumab[16] | A | A | A | A | A | – | – |
| PD-L1 | Atezolizumab[17] | A | – | – | A | – | – | – |
| | Avelumab[18] | – | – | – | – | – | – | – |
| | Durvalumab[19] | – | – | – | A | – | – | – |

'A' represents immune checkpoint inhibitors with at least one approved indication for the cancer type either as monotherapy or as part of combination therapy.

chemoradiotherapy[19] (online supplemental table 1). The populations in clinical trials tend to be selected based on prior evidence of antiproliferative or anti-tumour responses and favourable pharmacodynamics and pharmacokinetics in preclinical studies and early-stage trials.[22] For example, trials investigating the efficacy of anti-PD-L1 immune checkpoint inhibitors for ovarian cancer treatment have largely been restricted to evaluating patients with treatment-naïve advanced stage (stage III–IV) epithelial ovarian cancer, but have not been successful[23–27] (table 1, online supplemental table 1).

The interaction between PD-L1 on the surface of cancer cells and PD-1 on the surface of activated T cells suppresses further T cell activation.[10 20 21 28] Therefore, PD-1 and PD-L1 inhibitors suppress this interaction and so support T cell activation during anti-cancer immune responses.[10 12] While PD-1 is largely expressed on immune cells, PD-L1 is expressed on a wider variety of non-haematopoietic cells, including tumour cells.[29–32] Although there is uncertainty over the prognostic value of blood-based measures, previous studies have found that higher circulating (ie, blood-based) PD-1 and PD-L1 levels are associated with poorer prognosis for patients diagnosed with cancer at different anatomical sites. For example, higher plasma soluble PD-1 and PD-L1 expression levels were associated with decreased progression-free survival for patients with advanced-stage high-grade serous ovarian cancer compared with those with lower PD-1 and PD-L1 expression.[33] However, when accounting for other clinical factors in multivariable analyses, only soluble PD-L1 expression levels remained associated with progression-free survival for these patients.[33] Higher circulating soluble PD-L1 expression was also associated with decreased overall and progression-free survival in a meta-analysis of patients with cancer at different anatomical sites, including non-small cell lung cancer and melanoma patients who had been treated with immunotherapy.[34] In contrast, although low serum exosomal PD-L1 expression was associated with increased median overall survival for pancreatic ductal adenocarcinoma patients compared with those with high exosomal PD-L1 expression, there was little statistical evidence to support this observed difference.[35] Therefore, even though the prognostic roles of circulating PD-1 and PD-L1 expression levels have not been fully determined, there is some evidence supporting an association between blood-based measures of these immune checkpoint proteins and cancer survival, and the mechanism of action of these drugs is mediated by T cells.[33–35]

## Mendelian randomisation

Mendelian randomisation (MR) investigates the association between an exposure and outcome using genetic variants associated with the exposure as a proxy for the exposure of interest.[1] Two-sample MR uses measurements of genetic variant-exposure and genetic variant-outcome associations from separate studies, permitting analyses to leverage large-scale genetic association data for protein measures and cancer survival.[36 37]

MR should be less vulnerable to conventional issues of confounding, as genetic germline variants are randomly assorted at meiosis.[5 36–42] As germline genetic variants are fixed and cannot be influenced by subsequent disease status, MR analyses are immune to reverse causation bias.[5 36–40] Since MR analyses often use existing genetic association data, causal relationships can be tested in a more cost-effective and time-efficient manner than in randomised controlled trials.[1 36–38 40]

## Aims

The aim of this study is to investigate the association between genetically proxied circulating PD-1 and PD-L1 protein levels and survival of seven cancer types (breast, colorectal, head and neck, lung, melanoma, ovarian and prostate). These seven cancer sites have been chosen for inclusion as they have the most well-powered and accessible genome-wide association study (GWAS) survival data and make an important contribution to the overall global cancer burden.

These analyses will enable us to evaluate the repurposing potential of PD-1 and PD-L1 to new cancer indications. This will include potential repurposing to cancers without any existing approvals, as well as repurposing to new patient populations for cancers with some existing approvals. Previous MR studies have focused almost exclusively on causes of cancer risk. By including cancers with approved indications for PD-1 and PD-L1 inhibitors, which serves as a positive control, our analyses will also provide insight into the applicability of MR to studies of cancer survival.

## METHODS AND ANALYSIS
### Exposures

Rather than investigating the efficacy of specific immune checkpoint inhibitor compounds, their on-target effects will be proxied using genetic instruments which represent decreased circulating levels of their protein targets, PD-1 and PD-L1. Since our primary instruments will be based on studies in blood, we anticipate that our analyses may not fully proxy the mechanism of PD-1 and PD-L1 inhibitors in all biologically relevant tissues, an issue we will address in instrument validation analyses (see below).

Single-nucleotide polymorphisms (SNPs) associated with circulating PD-1 or PD-L1 expression levels will be used to proxy expression of these proteins. These genetic instruments will be selected from a GWAS of circulating proteins in 54219 participants of majority European ancestry in the UK Biobank cohort.[43] Statistical analysis, imputation, quality control and protein expression quantification in this study have been described previously.[43]

### Outcomes

Genetic association data will be obtained from GWAS of cancer survival in individuals of European ancestry with

**Table 2** Number of patients and mortality events occurring in the site-specific consortium genome-wide association study (GWAS) and Genomics England GWAS for each cancer site

| Cancer site | Number of patients | | | Number of events | | |
|---|---|---|---|---|---|---|
| | Consortium GWAS | Genomics England | Total | Consortium GWAS | Genomics England | Total |
| Breast | 91686[44] | 2183 | 93869 | 7531[44] | 238 | 7769 |
| Colorectal | 16964[45] | 2190 | 19154 | 4010[45] | 541 | 4551 |
| Head and neck | 10000‡ (unpublished) | 196 | 10196 | 3300 (unpublished) | 74 | 3374 |
| Lung | 7352 (unpublished) | 1318 | 8670 | 4598 (unpublished) | 592 | 5190 |
| Melanoma | 10982[46] | 219 | 11201 | 1041[46] | 108 | 1149 |
| Ovarian | 2901[47] | 494 | 3395 | 1656* | 217 | 1873 |
| Prostate | 24023[48] | −† | 24023 | 3513[48] | −† | 3513 |

The head and neck, lung and ovarian cancer mortality events are from all causes, while the mortality events for the other cancer sites are cancer site specific.
*The number of mortality events in the ovarian cancer GWAS was estimated based on the death proportion of 11 Ovarian Cancer Association Consortium (OCAC) studies (AUS, BAV, BEL, HAW, HSK, MAC, MAL, MAY, NCO, NEC, PVD) (0.571)[70] and the ovarian cancer GWAS sample size (2901).[47]
†There were fewer than 50 patients with prostate cancer included in the Genomics England survival GWAS so these will not be combined with the consortium site-specific prostate cancer GWAS.
‡The number of head and neck cancer cases in the consortium GWAS is an approximate estimate.

breast,[44] colorectal,[45] head and neck (unpublished), lung (unpublished), melanoma,[46] ovarian[47] and prostate cancer.[47] The outcome in each GWAS was defined as cancer-specific mortality, except for the lung and head and neck cancer GWAS which defined the outcomes as all-cause mortality, and the ovarian cancer GWAS which examined both progression-free survival and overall survival (all-cause) as outcomes.[44–48] To increase statistical power, we will combine the consortium site-specific GWAS with additional studies of cancer survival in Genomics England (unpublished) (table 2).

### Data harmonisation
Harmonisation of genetic data is the process by which the exposure and outcome GWAS summary statistics are joined together and oriented to reflect the same effect alleles. Therefore, harmonised data will only include SNPs which were common to both the protein expression and respective cancer survival GWAS.

Harmonisation will be performed using the harmonise_data function from the TwoSampleMR R package (https://mrcieu.github.io/TwoSampleMR/).[49 50] This will use the function's default option which infers the positive strand using allele frequencies for palindromic SNPs (https://mrcieu.github.io/TwoSampleMR/).[49 50] The correlation between exposure and outcome GWAS SNP effect allele frequencies will be compared following data harmonisation. If data harmonisation has been successful, the correlation coefficient would be expected to be close to 1, as this would suggest that the same alleles have been chosen as the effect allele in both GWAS summary statistic datasets.

### Mendelian randomisation
#### Assumptions
There are three key assumptions of MR: relevance, exchangeability and exclusion restriction.[39 42] The relevance assumption states that the genetic instrument must be associated with the exposure of interest, for example, in this study the circulating levels of the drugs' protein target.[36 42] The second MR assumption, exchangeability, requires there are no common causes of the instrument and outcome.[36 42] The final MR assumption, exclusion restriction, states that there must be no horizontal pleiotropy.[36 38–41 51] Horizontal pleiotropy occurs when there are additional pathways between the instruments and outcome, independent of the exposure.[36 42 51]

#### Genetic instrument selection
The UK Biobank protein expression GWAS summary statistics[43] will be used to select SNPs associated with circulating PD-1 or PD-L1 concentration. The genomic regions of the genes encoding PD-1 (*PDCD1*, chr2:241849884–241858894 in human genome build 38 (hg38)) and PD-L1 (*CD274*, chr9:5450503-5470566 in hg38) will be used to define *cis* and *trans* genetic instruments based on different window sizes.

To minimise vulnerability to horizontal pleiotropy, only SNPs within and in proximity to the gene encoding the target protein, known as *cis* SNPs, will be included in genetic instrument sets for the main analyses.[51] *Cis* instruments to proxy both proteins will be constructed in PLINK version 1.9[52 53] using SNPs in or within 500 kilobases (kb) from *PDCD1* or *CD274* that are associated with expression of these proteins ($p < 5 \times 10^{-6}$) at linkage disequilibrium (LD) $r^2 < 0.30$ (based on clumping with a random sample of 10000 European participants from the UK Biobank).[5 51 54]

### Estimator

Where the genetic instrument consists of one SNP, the Wald ratio will be used to assess the association between a protein instrumented by this SNP and cancer survival.[6 40 55] Where a genetic instrument consists of two or more SNPs, the inverse-variance weighted method will instead be used to investigate the association between a protein instrumented by these SNPs and cancer survival.[6 40 55] Any LD between SNPs included in an instrument will be accounted for in analysis using a SNP correlation matrix based on a random sample of 10 000 participants of European ancestry from the UK Biobank.[42 54 56] Heterogeneity of MR results across independent SNPs included in the genetic instrument sets will be assessed using Cochran's Q tests and MR results will be compared across each SNP in the instrument by visual inspection.[36 40 57]

The protein expression GWAS included age, age², sex, age–sex interaction terms, protein expression level measurement batch, UK Biobank centre, genotyping array, the first 20 principal components (PCs) of genetic ancestry, and duration between blood sample collection and protein expression measurement as covariates, in addition to preselection status of the participants for the replication cohort.[43]

Aside from the breast cancer GWAS which did not adjust for any covariates, the cancer site-specific GWAS were all adjusted for genetic PCs (although for different numbers of PCs).[44–48] The colorectal cancer survival GWAS additionally adjusted for age at diagnosis, sex, genotyping platform and study where the data originated from.[45] The head and neck cancer GWAS will be additionally adjusted for age, sex, stage at diagnosis (stratified as early or late stage) and cancer subtype. The lung cancer survival GWAS also adjusted for age and sex, and separated participants into early-stage (stage I–II), late-stage (stage III–IV) and all stage analyses (unpublished). The melanoma survival GWAS included age and sex as covariates in addition to genotyping batch for one cohort.[46] For the ovarian cancer survival GWAS, the primary study, residual disease, tumour stage, histology, tumour grade and age were also adjusted for.[47] The prostate cancer survival GWAS additionally included age, diagnostic prostate-specific antigen level and Gleason score as covariates.[48]

### Power

Using the UK Biobank protein expression GWAS,[43] the lead *cis* SNP for *PDCD1* (variant located at chr2:242801752 in human genome build 37 (hg37) with CG (A0) and C (A1) alleles) explained approximately 2.97% of the variation in circulating PD-1 expression level while the lead *cis* SNP for *CD274* (rs822341) explained approximately 4.17% of the variation in PD-L1 expression level.

Across all seven cancer types, there is an estimated power of 80% to detect HRs of at least ≥1.63 or ≤ 0.62 per unit decrease in normalised protein expression levels (alpha set to 5%) (table 3).

### Positive controls

Positive control analyses investigate the association between the exposure of interest and an outcome which

**Table 3** Estimated number of participants (N), death proportion, median survival, and HR per SD decrease detectable with 80% power for each cancer site

| Cancer | N | Death proportion | Median survival (months) | HRs detectable at estimated 80% power | |
|---|---|---|---|---|---|
| | | | | PD-1 | PD-L1 |
| Breast | 93 869 | 0.083 | 64.8[71] | HR ≥ 1.21<br>HR ≤ 0.83 | HR ≥ 1.17<br>HR ≤ 0.86 |
| Colorectal | 19 154 | 0.238 | 38.4[71] | HR ≥ 1.28<br>HR ≤ 0.78 | HR ≥ 1.22<br>HR ≤ 0.82 |
| Head and neck | 10 196 | 0.331* | 54.3[72] | HR ≥ 1.34<br>HR ≤ 0.75 | HR ≥ 1.26<br>HR ≤ 0.79 |
| Lung | 8670 | 0.599* | 3.6[71] | HR ≥ 1.27<br>HR ≤ 0.79 | HR ≥ 1.21<br>HR ≤ 0.83 |
| Melanoma | 11 201 | 0.103 | 53.4[73] | HR ≥ 1.63<br>HR ≤ 0.62 | HR ≥ 1.50<br>HR ≤ 0.67 |
| Ovarian | 3395 | 0.552* | 30.1[74] | HR ≥ 1.48<br>HR ≤ 0.68 | HR ≥ 1.37<br>HR ≤ 0.73 |
| Prostate | 24 023 | 0.146 | 62.4[71] | HR ≥ 1.33<br>HR ≤ 0.75 | HR ≥ 1.26<br>HR ≤ 0.79 |

HRs per standard deviation (SD) decrease estimated to be detected at 80% power calculated with the survSNP R package[75] (https://cran.r-project.org/web/packages/survSNP/index.html) using the combined estimated sample size and death proportion from each cancer survival genome-wide association study (GWAS) and the respective Genomics England cancer survival GWAS, median survival and assuming a false positive rate of 0.05. Death proportion was defined as the proportion of cancer cases who died due to all-cause or cancer-specific mortality. Transformation of protein level measurements into normalised protein expression (NPX) units on a log₂ scale have been described previously.[43]
*Death proportion due to all-cause mortality.

**Table 4** Number of incident UK Biobank cancer cases with protein expression data available for each cancer site and corresponding International Classification of Diseases 10th revision (ICD-10) code

| Cancer site | ICD-10 code | N |
|---|---|---|
| Breast | C50 | 780 |
| Colorectal | C18-20 | 545 |
| Head and neck | C00-C14, C32 | 124 |
| Lung | C34 | 391 |
| Malignant melanoma | C43 | 292 |
| Ovary | C56 | 89 |
| Prostate | C61 | 1154 |

Number of participants with each ICD-10 code obtained from Papier *et al* (2023).[62]

has already been observed to have a causal association with this exposure.[58] This enables the reliability of the genetic instruments for such exposures to be validated.[59] For these analyses, the positive control outcomes will be survival for cancers at sites which PD-1 or PD-L1 inhibitors have been approved for treatment by the MHRA (ie, breast, head and neck, colorectal, lung and melanoma cancer survival) (table 1). However, these analyses will be crude positive controls as these drugs are approved to treat highly specific patient populations (online supplemental table 1), whereas the cancer survival data have been generated from broader patient populations.

### Instrument validation

Our main analyses assume that SNPs associated with circulating PD-1 or PD-L1 protein expression level in the general population will have similar effects on protein levels in cancer cases and biologically relevant tissues (defined as those tissues responsible for the therapeutic benefit of PD-1 or PD-L1 inhibition). However, as PD-1 expression is upregulated due to T cell activation and PD-L1 expression is induced by inflammation and

carcinogenesis, SNP-protein effects may differ between the general population and cancer cases.[60 61] Thus, we will compare the strength and direction of SNP-protein associations among cancer cases, participants without a cancer diagnosis and in samples broadly representative of the general population in UK Biobank.[62] We will perform these analyses for UK Biobank cancer cases pooled across the seven cancer sites which our analyses focus on (n=3375), each of the seven cancer sites individually (table 4), and pooled across all cancer sites.[62]

The genetic instruments proxying circulating PD-1 or PD-L1 protein concentration will be further validated by investigating the strength and direction of the SNP-protein associations in UK Biobank cancer cases who were diagnosed with cancer prior to blood collection (prevalent cases) and those diagnosed with cancer following blood collection (incident cases). We will also assess whether the associations between these genetic instruments and circulating PD-1 or PD-L1 level differ by patient time since diagnosis. These sensitivity analyses will enable assessment as to whether the associations between these genetic instruments and circulating PD-1 or PD-L1 level (and so their strength as genetic instruments) differs for patients with cancer over time.

Additionally, as PD-L1 is expressed by cells at a number of potentially biologically relevant tissues aside from blood, such as the tumour site, endothelial cells and sites of metastases,[60] we will also investigate the association between the constructed genetic instruments and expression of the gene encoding PD-L1, *CD274,* in these tissues.

For each cancer site of interest, we will explore the strength and direction of association between these genetic instruments and *CD274* expression in tumour samples obtained from The Cancer Genome Atlas Program dataset (https://www.cancer.gov/tcga) (table 5) to validate the instruments' strength in the biologically relevant target population and target tissue.

We will also investigate the strength and direction of these associations for each anatomical site using tissue

**Table 5** Estimated number of tissue samples with germline genotype and gene expression data (N) available from The Cancer Genome Atlas Program (TCGA) dataset (https://www.cancer.gov/tcga) for each cancer site

| Cancer site | Study name | Study abbreviation | N |
|---|---|---|---|
| Breast | Breast invasive carcinoma | BRCA | 770 |
| Colorectal | Colon adenocarcinoma | COAD | 298 |
| | Rectum adenocarcinoma | READ | 109 |
| | Total | – | 407 |
| Head and neck | Head and neck squamous cell carcinoma | HNSC | 366 |
| Lung | Lung adenocarcinoma | LUAD | 385 |
| | Lung squamous cell carcinoma | LUSC | 334 |
| | Total | – | 719 |
| Melanoma | Skin cutaneous melanoma | SKCM | 54 |
| Ovarian | Ovarian serous cystadenocarcinoma | OV | 211 |
| Prostate | Prostate adenocarcinoma | PRAD | 366 |

Table 6   Estimated sample sizes (N) available for measurements of gene expression in tissues at each anatomical site of interest obtained from the Genotype-Tissue Expression (GTEx) version 8 dataset (https://www.gtexportal.org/home/tissue/)

| Tissue site | GTEx tissue name | N |
|---|---|---|
| Breast | Breast—mammary tissue | 396 |
| Colorectal | Colon—sigmoid | 318 |
| | Colon—transverse | 368 |
| | Total | 686 |
| Head and neck | Minor salivary gland | 144 |
| Lung | Lung | 515 |
| Melanoma | Skin—not sun exposed (suprapubic) | 517 |
| | Skin—sun exposed (lower leg) | 605 |
| Ovarian | Ovary | 167 |
| Prostate | Prostate | 221 |

sample data obtained from the Genotype-Tissue Expression database (https://www.gtexportal.org/home/) (table 6) and for immune cell populations obtained from Database of Immune Cell Expression, Expression quantitative trait loci and Epigenomics (DICE) (https://dice-database.org/). This will enable validation of the strength and direction of association of these instruments in the tissues of interest and further understanding of the background level of expression of this gene not specific to cancer biology.

Furthermore, as T cell level and function are affected by interactions between PD-1 and PD-L1, we will investigate the strength and direction of association of these genetic instruments with white blood cell count and function in general population samples as well as UK Biobank cancer cases and cancer-free participants.[10 12 60 63] This will serve as a positive control analysis, as we would expect that the genetic instruments proxying circulating PD-1 or PD-L1 levels will be associated with white blood cell metrics. Additionally, if the genetic instruments are associated with white blood cell metrics and cancer survival, this will provide evidence supporting a mechanism through which circulating PD-1 and PD-L1 level affects cancer survival mediated by white blood cells.

Overall, these validation analyses will enable assessment as to whether it is appropriate to assume that SNPs associated with circulating PD-1 or PD-L1 levels can also be used to proxy expression of these proteins at multiple anatomical sites in cancer cases and in biologically relevant tissues.

### Sensitivity analyses
The main analyses will be repeated using genetic instruments constructed with more stringent thresholds: significance p value thresholds of $< 5 \times 10^{-7}$ and $< 5 \times 10^{-8}$, window sizes of 250 kb and 100 kb on either side of the gene of interest, and LD $r^2$ thresholds of 0.2, 0.1 and

0.001. Although the primary analysis will only consider *cis* variants, *cis* and *trans* variants ($> 500$ kb from the gene of interest) will also be considered in secondary analyses. Instruments constructed from *cis* and *trans* variants will be selected based on p value and LD threshold ($p < 5 \times 10^{-8}$, $r^2 < 0.001$) with reference to a random sample of 10 000 participants of European ancestry from the UK Biobank. Where instruments are constructed from two or more SNPs, the primary analysis will also be re-run iteratively excluding individual SNPs from instruments to investigate whether findings are driven by individual SNPs.[42]

Colocalisation analysis will be performed using Pair-Wise Conditional analysis and Colocalisation analysis (https://github.com/jwr-git/pwcoco) to investigate whether any significant MR results are biased due to LD between one variant causing a change in protein expression and another causing a change in cancer survival through an independent pathway.[4 55 64 65]

Pleiotropy will be investigated by conducting phenome-wide association studies to investigate whether the genetic instruments proxying PD-1 or PD-L1 expression are also associated with other phenotypes. This will be achieved using the IEU Open GWAS project (https://gwas.mrcieu.ac.uk/).[49 50] The significance thresholds will be Bonferroni corrected for the number of traits looked up.[37 55] Although these methods will not specifically investigate horizontal pleiotropy, they will assess the possible extent of either vertical pleiotropy or horizontal pleiotropy. Vertical pleiotropy occurs when there is a mediator in the pathway between the exposure and outcome and, in contrast to horizontal pleiotropy, does not violate MR assumptions.[38 42 51 64]

Index event bias (also known as collider bias) may occur in studies of cancer survival if the hypothesised causal factor being evaluated for disease prognosis (in this case, PD-1 or PD-L1) is also a risk factor for disease onset.[4 37 66] If SNPs found to be significantly associated with cancer survival are also associated with risk of the same cancer type, methods including the SlopeHunter R package (https://github.com/Osmahmoud/SlopeHunter) will be used to evaluate and account for index event bias.[67 68]

### Secondary analyses
Where feasible, subgroup analyses will be performed to explore the impact of treatment history and tumour stage on the findings. We expect HRs to be larger in earlier tumour stages and in treatment-naive patients, compared with late-stage diagnoses and heavily treated patients, respectively.

### Software
The TwoSampleMR R package (https://mrcieu.github.io/TwoSampleMR/)[49 50] will be used to perform two-sample MR using summary-level data.

### Study status
Based on reviewer feedback during peer review of this protocol, we have added methods to the protocol to

evaluate the validity of constructed instruments, particularly with respect to potentially biologically relevant tissues other than blood,

Following submission of this protocol for publication, we have started to implement the analysis plan using breast, lung, melanoma, ovarian and prostate cancer survival as the outcomes of interest. We have also begun to perform sensitivity analyses detailed in this protocol, such as increasing the instrument construction criteria stringency and using cancer risk as the outcomes of interest, in addition to planned sub-group analyses for lung cancer survival across cancer stages. We anticipate completing these analyses in 2024 and submitting a paper detailing the findings of these analyses for publication by the end of 2024.

## Patient and public involvement

Members of an existing group of patients with cancer and caregivers volunteered to discuss this research project after a proposal had been drafted. The importance of potential side effects of medications and generalisability of findings to populations of non-European ancestry was highlighted. The concerns related to side effects will not be addressed here as they are outside of the scope of this analysis but will be considered as limitations and could be studied in the future using alternate data sources such as electronic medical record data. The analyses may be able to be performed for non-European populations if genetic survival data from patients with cancer of non-European ancestry are available. Patients and the public were not involved in the design of this study.

## Ethics and dissemination

No separate ethics approval will be required for these analyses as we will be using data from previously published genome-wide association studies which individually gained ethical approval and participant consent.

The protein expression GWAS conducted by Sun *et al* (2022) used UK Biobank data obtained under the approved application numbers 65851, 20361, 26041, 44257, 53639, 69804.[43]

The breast cancer survival GWAS conducted by Morra *et al* (2021) followed the Declaration of Helsinki principles.[44] The colorectal cancer survival GWAS conducted by Labadie *et al* (2022) gained approval by the Fred Hutchinson Cancer Research Center Institutional Review Board.[45] The head and neck cancer and lung cancer survival GWAS ethical approval information are unpublished. The melanoma survival GWAS conducted by Seviiri *et al* (2022) gained approval by the Sydney Local Health District Ethics Review Committee (MIA cohort), the United Kingdom's National North West Multi-Centre Research Ethics Committee (UK Biobank cohort) and the Human Research Ethics Committee of QIMR Berghofer Medical Research Institute (protocol).[46] The ovarian cancer survival GWAS conducted by Johnatty *et al* (2015) and the prostate cancer survival GWAS conducted by Szulkin *et al* (2015) included primary GWAS which

individually gained ethical approval from human research ethics committees.[47 48]

All participants involved in the protein expression GWAS and breast, colorectal, melanoma, ovarian (OCAC), and prostate cancer survival GWAS provided informed consent.[43–48 69]

The results of these analyses will be published and disseminated to members of the University of Bristol MRC Integrative Epidemiology Unit (IEU) Integrative Cancer Epidemiology Programme (ICEP) User Reference Group, which comprises patients with cancer and caregivers. The results will also be submitted as an open-access peer-reviewed publication and any statistical code will be made publicly available.

**Author affiliations**
[1]Medical Research Council Integrative Epidemiology Unit, University of Bristol, Bristol, UK
[2]Population Health Sciences, Bristol Medical School, University of Bristol, Bristol, UK
[3]University Hospitals Bristol and Weston NHS Foundation Trust and the University of Bristol, NIHR Bristol Biomedical Research Centre, Bristol, UK
[4]Department of Epidemiology and Biostatistics, School of Public Health, Imperial College London, London, UK

**Acknowledgements** We would like to acknowledge and thank the The International Lung Cancer Consortium (ILCCO) and the following ILCCO investigators for providing information on the lung cancer survival GWAS: Mei Dong, M. Catherine Brown, Rayjean J. Hung, Geoffrey Liu, Wei Xu. We would also like to acknowledge and thank Tom Dudding for providing information on the head and neck cancer survival GWAS.

**Contributors** JY and PH: conceptualisation, methodology, writing - review and editing, supervision. RM: writing - review and editing, supervision. TB: writing - original draft.

**Funding** TB is supported by grant MR/W006308/1 for the GW4 BIOMED MRC DTP, awarded to the Universities of Bath, Bristol, Cardiff and Exeter from the Medical Research Council (MRC)/UKRI. RMM is a National Institute for Health Research Senior Investigator (NIHR202411). RMM, JY and PCH are supported by a Cancer Research UK 25 (C18281/A29019) programme grant (the Integrative Cancer Epidemiology Programme). RMM is also supported by the NIHR Bristol Biomedical Research Centre which is funded by the NIHR (BRC-1215-20011) and is a partnership between University Hospitals Bristol and Weston NHS Foundation Trust and the University of Bristol. RMM is affiliated with the Medical Research Council Integrative Epidemiology Unit at the University of Bristol which is supported by the Medical Research Council (MC_UU_00011/1, MC_UU_00011/3, MC_UU_00011/6, and MC_UU_00011/4) and the University of Bristol. Department of Health and Social Care disclaimer: the views expressed are those of the author(s) and not necessarily those of the NHS, the NIHR or the Department of Health and Social Care. JY is also supported by the NIHR Imperial Biomedical Research Centre (BRC).

**Competing interests** RM, JY and PH have received funding from Cancer Research UK. TB receives funding from the GW4 BIOMED MRC DTP.

**Patient and public involvement** Patients and/or the public were not involved in the design, or conduct, or reporting, or dissemination plans of this research.

**Patient consent for publication** Not applicable.

**Provenance and peer review** Not commissioned; externally peer reviewed.

**Data availability statement** No data have been collected or generated as part of this protocol.

**ORCID iD**
Tessa Bate http://orcid.org/0009-0009-8034-2070

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
