## [Reviewer comments · BMJ Open]

ARTICLE DETAILS

TITLE (PROVISIONAL)	Investigating the association between genetically proxied circulating levels of immune checkpoint proteins and cancer survival: protocol for a Mendelian randomisation analysis
AUTHORS	Bate, Tessa; Martin, Richard; Yarmolinsky, James; Haycock, Philip

VERSION 1 – REVIEW

REVIEWER	Jiang, Xia Karolinska Institute
REVIEW RETURNED	02-Jul-2023

GENERAL COMMENTS	The authors plan to perform two-sample Mendelian randomisation to investigate the effect of genetically proxied expression of the protein targets of two immune checkpoint inhibitors programmed cell death protein 1 (PD-1) and programmed death ligand 1 (PD-L1) on survival of six cancer types (breast, colorectal, lung, melanoma, ovarian, and prostate), which would benefit cancer treatment through drug repositioning. The protocol is in general well-written. I have only a few minor issues. 1. As the authors stated, “across anti-PD-1 immune checkpoint inhibitors, there are approved indications for the treatment of breast, colorectal, and lung cancers and melanoma, whilst anti-PD-L1 immune checkpoint inhibitors have been approved for breast and lung cancer treatment” – this means that randomized clinical trials have already been conducted to test if these two immune checkpoint inhibitors also suit cancer treatment and some conclusions have, fortunately, been reached, i.e., they seems to work for breast, colorectal, lung cancers and melanoma. Why the authors took a step back and went to MR analysis to identify a putative causal relationship while several RCTs have already been perform for these two immune checkpoint inhibitors and cancer treatment? In other words, would the power of MR outperform RCTs to reach a firm conclusion? what are the add-on values of the current study? Maybe illustrate a little bit more of the shortages of RCTs, and the gap to be filled.2. the authors said, “analyses will enable study of the potential efficacy of these immune checkpoint inhibitors in broader cancer populations than typically investigated in clinical trials” – this might be technically true/appropriate but would perhaps in a price of losing clinical relevance. Specification is very important for cancer treatment; certain drug might only work within a few specific subtypes rather than a broad (or overall) cancer population.3. SNPs associated with circulating PD-1 or PD-L1 expression levels will be used to proxy expression of these proteins. Have the authors considered other tissues than blood?
--

	4. PD-1 inhibitors contains four drugs while PD-L1 inhibitor contains three drugs, but would these seven drugs make any difference when performing MR (i.e., different IVs for each of the seven drugs?), or would these seven drugs roughly fell into two broad categories (PD-1 or PD-L1)? In the latter case, why the authors bother to say “The approvals of anti-PD-1 and anti-PD-L1 immune checkpoint inhibitors are not uniform across cancer sites, even for the drugs targeting the same immune checkpoint protein” –since we are unable to distinguish the effects across seven drugs, knowing one or both of the immune checkpoint protein inhibitors work, that will be sufficient, isn’t? (and this loops back to my first comment)
--	--

REVIEWER	Lu, Lingeng Yale University
REVIEW RETURNED	21-Jul-2023

GENERAL COMMENTS	In this manuscript, the authors well described the rationale and their protocol (study plan and approaches) for the proposed study on PD-1 and PD-L1 and patient survival in six types of human cancer using two- sample mendelian randomization method. Some concerns I have for the authors to consider in revision.  1. In genetic instrument selection, the authors limited SNPs within the coding region for the genes of PD-1 and PD-L1 (page 8 line 14), and then within 500 kb from PD-1 or PD-L1 (line 18). Please clarify. Coding region only may be too conserved, and SNPs in downstream and upstream non-coding regions, promoter regions as well as intronic SNPs (e.g., PMID: 22822098) may also affect gene expression. 2. In the introduction, it would be expected to include some literature review on circulating PD-1 and PD-L1 and cancer patient survival (e.g. PMID: 37179293; 37127565;37004344, 36430974). 3. It would be greatly useful for readers if software and/or reference(s) for the power calculation in MR is provided.
---

VERSION 1 – AUTHOR RESPONSE

Responses to Reviewer 1:

Dr. Xia Jiang, Karolinska Institute

Comments to the Author:

The authors plan to perform two-sample Mendelian randomisation to investigate the effect of genetically proxied expression of the protein targets of two immune checkpoint inhibitors programmed cell death protein 1 (PD-1) and programmed death ligand 1 (PD-L1) on survival of six cancer types (breast, colorectal, lung, melanoma, ovarian, and prostate), which would benefit cancer treatment through drug repositing. The protocol is in general well-written. I have only a few minor issues.

1. As the authors stated, “across anti-PD-1 immune checkpoint inhibitors, there are approved indications for the treatment of breast, colorectal, and lung cancers and melanoma, whilst anti-PD-L1 immune checkpoint inhibitors have been approved for breast and lung cancer treatment” – this means that randomized clinical trials have already been conducted to test if these two immune checkpoint inhibitors also suit cancer treatment and some conclusions have, fortunately, been reached, i.e., they seems to work for breast, colorectal, lung cancers and melanoma. Why the authors took a step back and went to MR analysis to identify a putative causal relationship while several RCTs have already been perform for these two immune checkpoint inhibitors and cancer treatment? In other words,

would the power of MR outperform RCTs to reach a firm conclusion? what are the add-on values of the current study? Maybe illustrate a little bit more of the shortages of RCTs, and the gap to be filled. Response: The main advantage of performing MR for the four cancer types for which anti-PD-1 and/or anti-PD-L1 immune checkpoint inhibitors are approved is for those cancers to act as positive controls; i.e. if our genetic instruments are valid, we would expect decreased circulating PD-1 expression to be associated with decreased risk of lung cancer-specific death as the MHRA have approved PD-1 inhibitors in specific lung cancer populations. However, these positive controls are crude because, as mentioned in Comment 2, the current approvals are for highly specific patient populations (e.g. nivolumab is approved to treat metastatic or locally advanced non-small cell lung cancer patients who have previously been treated with chemotherapy) so will not match the less specific populations used in our analyses (i.e. stage I-IV lung cancer patients of European ancestry who have not been excluded based on treatment history).

We have added a 'Positive controls' section to the Methods and Analysis to clarify this in the section following where we describe MR: "Positive control analyses investigate the association between the exposure of interest and an outcome which has already been observed to have a causal association with this exposure (64). This enables the reliability of the genetic instruments for such exposures to be validated (65). For these analyses, the positive control outcomes will be survival for cancers at sites which PD-1 or PD-L1 inhibitors have been approved for treatment by the MHRA (i.e., breast, head and neck, colorectal, lung, and melanoma cancer survival) (Table 1). However, these analyses will be crude positive controls as these drugs are approved to treat highly specific patient populations (Supplementary Table 1), whereas the cancer survival data have been generated from broader patient populations." (page 9, line numbers: 21-30).

A second reason for evaluating the effect of immune checkpoint inhibitors for cancers where there are already approvals for specific patient populations is to explore whether the benefits of these medications could be expanded to broader patient groups (further discussed in response to Comment 2).

2. The authors said, "analyses will enable study of the potential efficacy of these immune checkpoint inhibitors in broader cancer populations than typically investigated in clinical trials" – this might be technically true/appropriate but would perhaps in a price of losing clinical relevance. Specification is very important for cancer treatment; certain drug might only work within a few specific subtypes rather than a broad (or overall) cancer population.

Response: We aim to increase the specificity of patient populations used in our analyses to some extent by restricting populations based on treatment history and tumour stage status in the secondary analyses, where possible (page 12), as these are two characteristics which contribute to the definition of some patient subgroups approved for immune checkpoint inhibitor treatment.

3. SNPs associated with circulating PD-1 or PD-L1 expression levels will be used to proxy expression of these proteins. Have the authors considered other tissues than blood?

Response: We have chosen blood as the tissue for PD-1/L1 expression level measurements as immune checkpoint inhibitors promote systemic immune responses and because large genome-wide associations studies of proteins in other tissues are unavailable. Supporting the biological relevance of blood, the therapeutic benefit of PD-1 inhibition is mediated by white blood cells and there is also some evidence that circulating levels of PD-1/L1 are associated with cancer survival (further discussed in response to Reviewer 2 Comment 2). However, we acknowledge that these may not fully proxy the mechanism of action of these immune checkpoint inhibitor drugs due to the expression of both of these proteins (particularly PD-L1) in a range of potentially biologically relevant tissues other than blood.

To address this limitation further, we intend to implement further methods to validate these instruments, as detailed in the 'Instrument Validation' section which has been added to the protocol. These validation methods include investigating whether PD-L1 genetic instruments are associated with CD274 (gene encoding PD-L1) expression in tumour samples using The Cancer Genome Atlas Program (TCGA) dataset, and in multiple biologically relevant tissue sites in the Genotype-Tissue

Expression (GTEx) and Database of Immune Cell Expression, Expression quantitative trait loci (eQTLs) and Epigenomics (DICE) datasets.

4. PD-1 inhibitors contains four drugs while PD-L1 inhibitor contains three drugs, but would these seven drugs make any difference when performing MR (i.e., different IVs for each of the seven drugs?), or would these seven drugs roughly fell into two broad categories (PD-1 or PD-L1)? In the latter case, why the authors bother to say “The approvals of anti-PD-1 and anti-PD-L1 immune checkpoint inhibitors are not uniform across cancer sites, even for the drugs targeting the same immune checkpoint protein” –since we are unable to distinguish the effects across seven drugs, knowing one or both of the immune checkpoint protein inhibitors work, that will be sufficient, isn’t? (and this loops back to my first comment)

Response: We will be using the two categories of drug targets (PD-1 or PD-L1) as the exposures of interest, rather than the seven individual drugs. We agree with the reviewer that the effects of immune checkpoint inhibitors targeting the same protein cannot be distinguished between using MR and have solely included the information in the sentence highlighted by the reviewer as background information on the current indications of these drugs. Differences in the approvals of immune checkpoint inhibitors targeting the same proteins may be due to differences in off-target effects of the drugs which are not examined in MR or differences in clinical trials providing approvals for these drugs (i.e. due to differences in patient populations, or lack of a late-stage clinical trial rather than failure of a performed clinical trial).

Responses to Reviewer 2:

Dr. Lingeng. Lu, Yale University

Comments to the Author:

In this manuscript, the authors well described the rationale and their protocol (study plan and approaches) for the proposed study on PD-1 and PD-L1 and patient survival in six types of human cancer using two- sample mendelian randomization method. Some concerns I have for the authors to consider in revision.

1. In genetic instrument selection, the authors limited SNPs within the coding region for the genes of PD-1 and PD-L1 (page 8 line 14), and then within 500 kb from PD-1 or PD-L1 (line 18). Please clarify. Coding region only may be too conserved, and SNPs in downstream and upstream non-coding regions, promoter regions as well as intronic SNPs (e.g., PMID: 22822098) may also affect gene expression.

Response: The cis genetic instruments will be constructed using window sizes of 500 kb (main analyses) (page 8), 250 kb, or 100 kb (the latter two as sensitivity analyses) (page 9). We have modified the sentence to clarify that the coding region in addition to variable window sizes will be used to construct the genetic instruments: “The genomic regions of the genes encoding PD-1 (PDCD1, chr2:241849884 – 241858894 in human genome build 38 (hg38)) and PD-L1 (CD274, chr9:5450503 – 5470566 in hg38) will be used to define cis and trans genetic instruments based on different window sizes.” (page 8, line numbers: 8-11).

2. In the introduction, it would be expected to include some literature review on circulating PD-1 and PD-L1 and cancer patient survival (e.g. PMID: 37179293; 37127565;37004344, 36430974).

Response: We have added a section (page 5, line numbers: 6-24) summarising the association between increased circulating PD-1 and PD-L1 expression levels and poor prognosis based on these suggested papers:

“Although there is uncertainty over the prognostic value of blood-based measures, previous studies have found that higher circulating (i.e. blood-based) PD-1 and PD-L1 levels are associated with poorer prognosis for patients diagnosed with cancer at different anatomical sites. For example, higher plasma soluble PD-1 and PD-L1 expression levels were associated with decreased progression-free survival for patients with advanced-stage high-grade serous ovarian cancer compared to those with lower PD-1 and PD-L1 expression (33). However, when accounting for other clinical factors in multivariable analyses, only soluble PD-L1 expression levels remained associated with progression-

free survival for these patients (33). Higher circulating soluble PD-L1 expression was also associated with decreased overall and progression-free survival in a meta-analysis of patients with cancer at different anatomical sites, including non-small cell lung cancer and melanoma patients who had been treated with immunotherapy (34). In contrast, although low serum exosomal PD-L1 expression was associated with increased median overall survival for pancreatic ductal adenocarcinoma patients compared to those with high exosomal PD-L1 expression, there was little statistical evidence to support this observed difference (35). Therefore, even though the prognostic roles of circulating PD-1 and PD-L1 expression levels have not been fully determined, there is some evidence supporting an association between blood-based measures of these immune checkpoint proteins and cancer survival, and the mechanism of action of these drugs is mediated by T cells (33-35).”

3. It would be greatly useful for readers if software and/or reference(s) for the power calculation in MR is provided.

Response: The power calculations were performed using the survSNP package in R which was cited in the footer of Table 3 (page 9). A web link to the CRAN page for the survSNP package has been added to the footer of Table 3 to increase clarity for readers (page 9, line 16-17).

Further modifications

We have added head and neck cancer survival as an outcome of interest for the MR analyses and so the protocol has been revised throughout to reflect this.

We have also added a section detailing ‘Instrument Validation’ to the Methods and Analysis.

VERSION 2 – REVIEW

REVIEWER	Jiang, Xia Karolinska Institute
REVIEW RETURNED	14-Oct-2023

GENERAL COMMENTS	The authors have answered my questions properly.
--

REVIEWER	Lu, Lingeng Yale University
REVIEW RETURNED	21-Oct-2023

GENERAL COMMENTS	The authors have appropriately addressed the concerns, and no more comments are raised.
---

VERSION 2 – AUTHOR RESPONSE